# Adherence to the test, treat and track strategy for malaria control among prescribers, Mfantseman Municipality, Central Region, Ghana

Ernestina Esinam Agbemafle[1,2☯], Adolphina Addo-Lartey[3‡], Magdalene Akos Odikro [2,3‡]*, Joseph Asamoah Frimpong[2‡], Chrysantus Kubio[4☯], Donne Kofi Ameme[2‡], Samuel Oko Sackey[2,3‡], Harriet Affran Bonful[3‡]

1 Mercy Women's Catholic Hospital, Mankessim, Ghana, 2 Ghana Field Epidemiology and Laboratory Training Programme, Department of Epidemiology and Disease Control, School of Public Health, College of Health and Allied Sciences, University of Ghana, Legon, Ghana, 3 Department of Epidemiology and Disease Control, School of Public Health, College of Health and Allied Sciences, University of Ghana, Legon, Ghana, 4 Savannah Regional Health Directorate, Ghana Health Service, Damongo, Ghana

☯ These authors contributed equally to this work.
‡ AAL, MAO, JAF, DKA, SOS and HAB also contributed equally to this work.
* odikrom@gmail.com

## Abstract

### Background

The test, treat, and track (T3) strategy is directed at ensuring diagnosis and prompt treatment of uncomplicated malaria cases. Adherence to T3 strategy reduces wrong treatment and prevents delays in treating the actual cause of fever that may otherwise lead to complications or death. Data on adherence to all three aspects of the T3 strategy is sparse with previous studies focusing on the testing and treatment aspects. We determined adherence to the T3 strategy and associated factors in the Mfantseman Municipality of Ghana.

### Methods

We conducted a health facility based cross-sectional survey in Saltpond Municipal Hospital and Mercy Women's Catholic Hospitals in Mfantseman Municipality of the Central Region, Ghana in 2020. We retrieved electronic records of febrile outpatients and extracted the testing, treatment and tracking variables. Prescribers were interviewed on factors associated with adherence using a semi-structured questionnaire. Data analyses was done using descriptive statistics, bivariate, and multiple logistic regression.

### Results

Of 414 febrile outpatient records analyzed, 47 (11.3%) were under five years old. About 180 (43.5%) were tested with 138 (76.7%) testing positive. All positive cases received antimalarials and 127 (92.0%) were reviewed after treatment. Of 414 febrile patients, 127 (30.7%) were treated according to the T3 strategy. Higher odds of adherence to T3 were observed for patients aged 5–25 years compared to older patients (AOR: 2.5, 95% CI: 1.27–4.87,

**Data Availability Statement:** All relevant data are within the paper and its Supporting Information files.

**Funding:** The author(s) received no specific funding for this work.

**Competing interests:** The authors have declared that no competing interests exist

p = 0.008). Adherence was low among physician assistants compared to medical officers (AOR 0.004, 95% CI 0.004–0.02, p<0.001). Prescribers trained on T3 had higher adherence (AOR: 99.33 95% CI: 19.53–505.13, p<0.000).

## Conclusion

Adherence to T3 strategy is low in Mfantseman Municipality of the Central Region of Ghana. Health facilities should perform RDTs for febrile patients at the OPD with priority on low cadre prescribers during the planning and implementation of interventions to improve T3 adherence at the facility level.

## Introduction

Malaria causes high morbidity and mortality, especially among children under five years. Although malaria mortality has been reduced by over a quarter around the world, and by one third in WHO African Region, its transmission still occurs in 99 countries [1]. In 2020, an estimated 241 million malaria cases and 627,000 malaria deaths occurred worldwide and children under five years were mostly affected [2]. Ensuring adequate treatment of suspected malaria cases is a key approach to malaria control [3].

The test, treat and track (T3) strategy for malaria was introduced by the WHO in 2010. The T3 strategy is an initiative that supports malaria-endemic countries in their effort to achieve universal coverage with diagnostic testing, anti-malarial treatment and surveillance. It also seeks to update existing malaria control and elimination strategy, as well as country-specific operational plans [4]. Ghana subscribed to sub-regional and global initiative of T3 in 2013 [3]. Ghana's National Malaria Control Program (NMCP) subsequently developed guidelines for the implementation of the T3 strategy [3]. This initiative seeks to ensure that every suspected malaria case is tested, that every case tested positive is treated with the recommended quality-assured antimalarial medicines, and that the disease is tracked through timely and accurate reporting to guide policy and operational decision [5]. These processes if strictly adhered to will enhance an accurate profiling of malaria burden and also contribute to appropriately managing other causes of febrile illnesses [6]. It will additionally reduce the unnecessary exposure of patients to anti-malarial medicines, reduce consumption of ACTs and thus eliminate pressure on medicines [7]. Overall, adherence to the T3 strategy is critical to Ghana's goals of controlling and subsequently eliminating Malaria in the country.

However, the practice of presumptive treatment for malaria continues to persists across the country with low adherence to the T3 strategy in health facilities. Although the proportion of OPD malaria cases that were tested has increased from 39% in 2013 to 78% in 2016 [3], presumptive treatment and prescription of antimalarials to persons with negative tests are still prevalent. Findings from a cross sectional study conducted in the Bongo District in the Northern Region of Ghana indicated that adherence to T3 at CHPS Compound was 60.5%, 30.1% in Health Centres, and 39.5% in Hospitals [8]. Some studies have shown that frequent stock-out of RDT, lack of diagnostic facilities, unavailability of laboratory resources, ACT stock-outs, absence of monitoring and supervision amongst others are the major challenges that hinder the adherence to the T3 strategy [8, 9].

Two main deviations exist in the implementation of the T3 strategy [10–14]. These deviations include: failure of prescribers to test all febrile patients prior to the application of antimalaria medications and also, issues with the tracking component of the strategy (review of all patients treated with antimalarial medications). Several studies have been conducted on the

adherence to the T3 strategy in other parts of the country [8, 9, 15, 16]. However, these studies have concentrated mainly on the testing and treatment aspects of the T3 strategy. In most cases it has been reported that afebrile patients were treated without malaria testing [17–19]. Few studies have focused on the tracking component of the strategy with one study conducted in Bongo District reporting that less than half of patients treated were followed-up or reviewed [8].

The limited information on the adherence to the strategy in the Mfantseman Municipality of the Central Region of Ghana and most especially on the third component (tracking) calls for evaluation of the implementation of the T3 strategy in this area. We therefore sought to find out the level of adherence to the T3 strategy and also assessed the factors that are associated with prescriber adherence to T3 in the municipality.

## Methods

### Study design

This is a hospital-based cross-sectional study carried out in two hospitals in the Mfantseman Municipality, from November 2019 to February 2020. The assessment was done at two distinct levels: to identify the exposure variables, we interviewed prescribers and determined the outcome variables by reviewing records of OPD patients who were seen by the prescribers and treated for uncomplicated malaria.

### Study location and setting

We carried out the study in the Mfantseman Municipality of the Central Region of Ghana. Mfantseman Municipal is located along the coastline of the Central Region. The 2020 estimated population of the municipality is 185,554 with a population growth rate of 2.8% [20]. The municipality has 32 public and 4 private health facilities. These include two hospitals, five health centers, twenty-four Community Health Planning and Services (CHPS) Compounds, four clinics. In addition, there are four other private health facilities and several Chemical shops and Diagnostic centers. The study was carried out in two secondary level health facilities namely, Saltpond Municipal Hospital (SMH) which is located at the Municipal capital Saltpond, and Mercy Women's Catholic Hospital (MWCH) located in the largest commercial city in the municipality, Mankessim. These health facilities were purposively selected because they are the main referral centers for the municipality.

### Study population

The study population included patients who visited the out-patient department and prescribers who manage outpatients at the study site. Study subjects included patients who reported to the outpatients' department with a history of fever and were suspected of having malaria during the study period and prescribers including clinicians involved in managing suspected malaria cases. Patients visiting the health facility for follow-up for life-long illnesses such as diabetes, hypertension, burns, trauma, patients on admission and clinicians who were on leave during the period of data collection were excluded from the study.

### Operation of the health care system and background of prescribers

Generally, for every patient that visits the hospital to seek health care, the first point of contact is with records department where patient particulars are given to the records officers. The next point of call is the triage area where nurses assess the vital signs such as temperature, pulse, respiration, blood pressure as well as the oxygen saturation of the patient. The patient is then sent

to the consulting room to be seen by a prescriber. Patients are referred to the laboratory for testing, when necessary, in this context, malaria testing is done in two ways; either by microscopy or by the use of the rapid diagnostic testing or both. Patients goes to the prescriber again with their results for treatment per their test results and they receive their medication at the pharmacy/dispensary after which they make payments at the billing and revenue section of the hospital based on their insurance status. Patients who have an insurance cover provide their insurance cards and do not make payment while uninsured patients pay for their services.

Prescribers are mainly medical doctors, physician assistants and nurse prescribers. In Ghana, Medical doctors go through a six to seven-year training at the university and two years of house job after which they are awarded an MBChB-Bachelor of Medicine & Surgery. Physician assistants either go through a four-year training in the university after which they do a one-year internship of clinical practice and are awarded a BSc Physician Assistantship degree or a nurse goes through the physician assistantship training for two years after at least two years of practicing as a nurse. The nurse prescriber is a nurse or midwife who is either trained on the job to attend to patients on OPD basis or underdoes a two-year training to become a nurse prescriber. Although these prescribers belong to different professional categories and their competences vary widely, they are all expected to adhere to the T3 strategy. All these categories of professionals are expected to be trained on the T3 strategy either through orientation or in-service training.

## Sampling and sample size calculation

The minimum sample size was estimated using the Cochran's sample size formula: $n_o = Z^2pq/e^2$ [21]. Where; $n_o$ is minimum estimated sample size, Z is the z score corresponding to the chosen alpha level of 0.5 (95% confidence interval) which is 1.96, p is the estimated proportion of prescriptions conforming to the T3 guideline which is 42.5% from previous study, q is 1-p and e is the margin of error–Bartlett et al recommend using 5% [22]. We estimated the percentage of prescriptions that comply to the T3 guideline using the 42.5% adherence from a previous study by Akanteele Agandaa et al [8]. Substituting the values into the equation, the sample size was estimated as $(1.96^2 * 0.425 * (1–0.425)/ 0.05^2 = 376$. A 10% allowance for lapses that might occur during data collection was assumed, hence 414 patient records were targeted for the study. All prescribers in the two study sites were eligible for interviews.

To determine the distribution of the calculated sample size across the two study sites, the electronic data of patients who were treated for malaria for the preceding one year to the commencement of the study was used. A monthly average of about 603 attendants diagnosed as malaria was found at the two sites. Of these, MWCH had 394 (65.3%) and SMH had 209 (34.7%). Using the probability proportionate to size method, 144 records were reviewed from SMH and 270 from MWCH to achieve our sample size. The targeted total of 414 records was reviewed in the study using systematic random sampling. At MWCH, an average of 14 patients were diagnosed with malaria each day, since 270 records were targeted, a sampling interval of 2 was used. The first record was selected using random number generator and with an interval of 2, Seven records were reviewed on each day for four months. At SMH, an average of 7 patients were diagnosed with malaria each day. A sampling interval of 2 was used. The first sample was selected using a random number generator and with the interval of 2, 4 records were reviewed on each day for four months.

## Data collection

Data was collected from November 2019 to February 2020. Data were obtained from two main sources: electronic patient database and prescribers. Following a pre-test, a checklist was used

to abstract data on patient age, sex, date of attendance, and insurance status. Data were collected on whether suspected malaria cases were tested before treatment and whether the prescriber indicated that the patient should return for follow-up or review. Interviews were conducted with prescribers in both health facilities to obtain data comprising socio-demographic characteristics of prescribers. Prescriber knowledge of T3 strategy including training on the strategy, understanding of the T3 strategy, and the reasons for using the strategy, health facility factors such as number of qualified laboratory scientist available and NMCP monitoring were also assessed. The primary outcome variable, adherence to the T3 strategy: is a composite variable that combines testing, treating, and tracking of malaria. It was derived as an addition of the three components of the strategy i.e., Testing +Treating + Tracking = T3 Adherence. To meet the criteria for the outcome variable, the prescriber should have requested for microscopy or RDT for any suspected malaria case, treat confirmed malaria cases with the recommended antimalarial, and asked the patient to return for a review. If all these were done, it implied prescriber adhered to the T3 strategy. The outcome variable was coded as a binary variable i.e., adherence and non-adherence. Independent variables included patient factors such as age, sex, and insurance status and prescriber factors. Prescriber factors included socio-demographic characteristics of clinicians such as age, sex, professional category, years of experience, and training on T3 strategy. Health facility factors included; availability of functional laboratory, the number of qualified laboratory scientists. Other external factors such as policies such as NHIA policy on T3 and NMCP monitoring were assessed.

## Data processing and analysis

Data entry was done with Microsoft excel 2013 and SPSS version 22 was used for coding. The data was then exported to STATA/SE version 15 for analysis. The two data sets, i.e., record review and interview of prescribers were merged using prescribers as the principal identity (ID). The data were declared as a survey data before analysis using the svy command in STATA/SE version 15 thereby adjusting for possible clustering at the health facility. Summary descriptive statistics were conducted and presented as frequencies and proportions, in tables and graphs. Univariate analysis was performed to determine the crude association between outcome variables and other predictor variables using odds ratios and 95% confidence intervals (CIs). A p-value of less than 0.05 was considered significant. In determining a combination of patient, health facility, and prescriber factors that are associated with adherence, the outcome variables and all the exposure variables that predicted the outcome at p<0.1 in the crude analysis were placed in a multiple logistic regression model. These variables included prescriber age, sex, professional category, number of years in service, training, last training in T3 strategy, NMCP monitoring and patient age. Associations were considered significant at P-Value of 0.05 or less.

## Ethical approval and consent to participate

Approval for this study was obtained from the Ghana Health Service Ethics Review with approval number GHS-ERC045/11/19. Formal permission was sought from the Regional Health Directorate, Mfantseman Municipal Health Information Management Team (MHMT) and the Medical Directors of the two hospitals involved in the study before data collection. All information concerning individual patients were fully anonymized in the data set. Written consent was also obtained from prescribers before questionnaires were administered. All data generated from this study was password protected and only accessible by study team members.

## Results

### Descriptive characteristics of respondents

Of the 414 febrile outpatients, 47 (11.3%) were less than 5 years old. Their ages ranged from 7 months to 89 years with a median age of 27 years. Females were 383 (68.1%) of the patients. Additionally, 375 (90.6%) of the patients were registered on the National Health Insurance Scheme (NHIS). These patients were attended to by eighteen prescribers, whose mean age was 32.1 years with standard deviation (SD) ± 2.74 years. There were six medical officers, six physician assistants, and six nurse prescribers. The mean number of years in the practice of the prescribers was 4.9 years (SD) ± 3.3 years.

With regards to prescriber-patient interaction, 184 (44.4%) of patients were attended to by physician assistants, 108 (26.1%) were seen by medical officers while the remaining 122 (29.6%) were seen by nurse prescribers (Table 1).

### Malaria testing, treatment, and tracking patterns of prescribers

Of the 414 febrile patients who were treated for malaria, 180 (43.5%) of them were tested of which 161 (89.4%) were done by microscopy and 19 (10.6%) by RDTs. Of the180 tests conducted, 138 (76.7%) were positive. All the 414 febrile patients were prescribed antimalarials irrespective of whether they were tested or not, and whether the result was positive or negative for those tested. All the 138 (100%) patients who tested positive for malaria, were prescribed the recommended ACTs. Of the 138 confirmed malaria case-patients who were treated in the municipality, 127 (92.0%) were asked to return to the health facility for follow-up review between 5 to 7 days. The proportion of the 414 febrile patients who were treated with or without testing, 276 (66.7%) were tracked. Overall, the proportion of febrile patients who were treated according to the T3 strategy for malaria control was 30.7% (127/414) (Fig 1).

### Factors associated with adherence to the T3 for malaria control

Prescribers had 2.5 times odds of adherence when attending to patients aged between five to twenty-four years and 3.1times odds of adherence when attending to patients aged twenty-five to forty-four years compared to children four years or less (AOR: 2.5, 95% CI: 1.27–4.87, p = 0.008) and (AOR: 3.1, 95% CI: 1.49–6.33.0, p = 0.002) respectively. Similarly, there was no change in odds of prescriber adherence with regards to the patient's insurance status (AOR: 1.06, 95% CI: 0.52–2.15, p = 0.882). For health facility level factors, prescribers had 2.9 times odds of adherence with 3 qualified laboratory scientists in a health facility compared with less than 3 laboratory scientists (AOR: 2.90, 95% CI: 1.31–6.19, p = 0.008). The odds of adherence were 0.64 times lower among prescribers monitored by the NMCP compared to those who have not been monitored by the NMCP (AOR: 0.36, 95% CI: 0.21–0.62), p<0.000.

With regard to prescriber level factors, as prescribers age, their odds of adherence to the T3 strategy changes significantly by 0.38 times lower each year compared to the previous year's adherence (AOR 0.62, 95% CI: 0.49–77, p<0.000). Female prescribers had 13.56 times higher odds of adherence compared to their male counterparts (AOR: 13.56, 95% CI: 1.64–111.9, p<0.015). Physician assistants had 0.99 times lower odds compared to that of medical officers (AOR 0.004 95% CI: 0.004–0.02, p<0.001). Prescribers' who have worked five years and more had 0.3 times lower odds of adherence compared to those who have worked less than five years (AOR: 0.69, 95% CI: 0.60–0.78, p<000). Also trained prescribers had 98.3 times higher odds of adherence compared to untrained prescribers (AOR: 99.33 95% CI: 19.53–505.13, p<0.000). Prescribers who have been trained for more than 6 months on the T3 strategy had

**Table 1. Demographic characteristics of study participants.**

| Characteristic | Frequency(N = 414) | Percent (%) |
|---|---|---|
| **a. Health facility** | | |
| MWCH | 270 | 65.2 |
| SMH | 144 | 34.8 |
| **b. Patient** | | |
| **Sex** | | |
| Male | 132 | 31.9 |
| Female | 282 | 68.1 |
| **Age group** | | |
| Under 5 years | 47 | 11.4 |
| 5–24 | 150 | 36.2 |
| 25–44 | 113 | 27.3 |
| 45–65 | 64 | 15.5 |
| 65+ | 40 | 9.7 |
| **Insurance status** | | |
| Insured | 375 | 90.6 |
| Not insured | 39 | 9.4 |
| **c. Prescriber** | | |
| **Sex** | | |
| Male | 13 | 68.4 |
| Female | 6 | 31.6 |
| **Age group** | | |
| 30 or less | 8 | 42.1 |
| 31–35 | 7 | 36.8 |
| 36+ | 4 | 21.1 |
| **Professional category** | | |
| Medical officer | 6 | 31.6 |
| Physician assistant | 6 | 31.6 |
| Nurse prescriber | 7 | 36.8 |
| **Years in service** | | |
| 5 years or less | 13 | 68.4 |
| 6 years or more | 6 | 31.6 |
| **Training on T3 strategy** | | |
| Yes | 16 | 84.2 |
| No | 3 | 15.8 |
| **Last training** | | |
| 1 year or less | 13 | 68.4 |
| 2 years or more | 3 | 15.8 |
| No training | 3 | 15.8 |

7.4 times odds of adherence compared to those who were trained less than 6 months ago (AOR 7.43 95% CI 4.62–11.95, p<0.000) (Table 2).

## Discussion

We sought to determine the level of adherence to the T3 strategy of and its associated factors. In our study, findings revealed that of expected 100% adherence, T3 was adhered to for less than half of the patients. Factors that were associated with the adherence to the T3 strategy

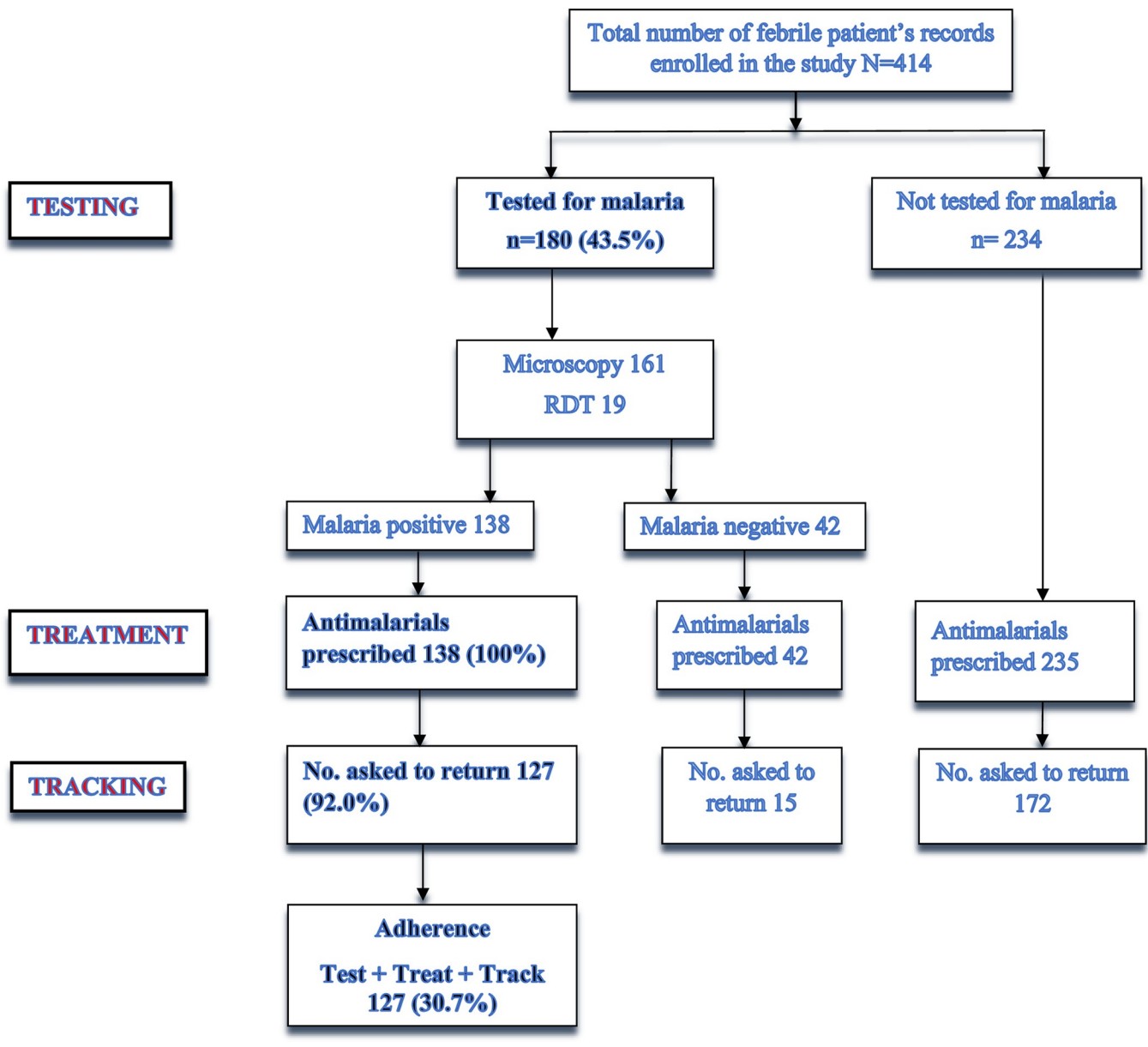

**Fig 1.**

included patient age, number of qualified laboratory scientists in a facility, NMCP monitoring, prescriber age and sex, professional category, number of years in service and training.

It was observed that less than half of suspected malaria case patients were either tested by microscopy or RDT. Studies in other countries have reported varying percentages of testing of suspected malaria cases. Compared to the findings from this study, higher testing of suspected malaria cases were reported in other studies across the continent which ranged from 64.6% to 85.2% [6, 13, 23–25]. In Ogun State Nigeria, 85.2% testing rate was established in public facilities while 32.9% testing was done in private facilities [26]. On the other hand, lower testing of suspected malaria cases have been reported in other studies in sub-Saharan Africa, Malawi 34.0% [27]. Similar findings to the testing rate in our study, are studies in Sudan 43.5% [11],

**Table 2. Multivariate analysis of association between patient, health facility and prescriber related variables and adherence to T3 strategy, Mfantseman Municipality, 2022.**

| Variable | Adherence to T3 strategy (N = 414) | | Unadjusted | p-value | Adjusted | p-value |
|---|---|---|---|---|---|---|
| | | | OR (95% CI) | | OR (95% CI) | |
| | Yes | No | | | | |
| | N (%) | N (%) | | | | |
| **a. Patient-related variables and adherence to T3 strategy** | | | | | | |
| Age | | | | | | |
| ≤4 | 23(5.6) | 23(5.6) | 1 | | 1 | |
| 5–24 | 44(10.6) | 106(25.6) | 1.09(0.55–2.17) | 0.807 | 2.49(1.27–4.87) | 0.008 |
| 25–44 | 28(6.8) | 85(20.5) | 1.02(0.49–2.09) | 0.954 | 3.08(1.49–6.33) | 0.002 |
| 45–64 | 24(5.8) | 40(9.7) | 0.76(0.34–1.71) | 0.504 | 1.68(0.77–3.64) | 0.188 |
| 65+ | 13(3.1) | 27(6.5) | 1.04(0.43–2.53) | 0.925 | 2.12(0.88–5.10) | 0.094 |
| Sex | | | | | | |
| Male | 86(30.7) | 46(34.6) | 1 | | 1 | |
| Female | 195(69.6) | 87(65.4) | 1.19 (0.77–1.85) | 0.417 | 1.13(0.72–1.78) | 0.586 |
| Insurance status | | | | | | |
| Not insured | 26(6.3) | 13(3.1) | 1 | | 1 | |
| Insured | 255(61.6) | 13(3.1) | 1.06(0.53–2.14) | 0.865 | 1.06(0.52–2.15) | 0.882 |
| **b. Health facility-level variables and adherence to T3 strategy** | | | | | | |
| Number of qualified laboratory scientists | | | | | | |
| ≤3 | 23(5.6) | 15(3.6) | 1 | | 1 | |
| >3 | 258(62.3) | 23(5.6) | 1.43(0.72–2.83) | 0.311 | 2.9(1.31–6.19) | **0.008** |
| NMCP Monitoring | | | | | | |
| No | 132(31.9) | 1(0.2) | 1 | | 1 | |
| Yes | 161(38.9) | 120(29.0) | 0.34(0.22–0.55) | 0.000 | 0.36(0.21–0.62) | **0.000** |
| **c. Prescriber-related variables and adherence to T3 strategy** | | | | | | |
| Age | | | 0.82(0.75–0.88) | 0.000 | 0.62(0.49–77) | **0.000** |
| Sex | | | | | | |
| Male | 245(51.2) | 130(31.4) | 1 | | 1 | |
| Female | 36(8.7) | 3(0.7) | 6.37(1.92–21.07) | 0.002 | 13.56(1.64–111.97) | 0.015 |
| Professional category | | | | | | |
| Medical officer | 108(26.1) | 0 | 1 | | 1 | |
| Physician assistant | 166(40.1) | 18(4.3) | 151.51(61.30–374.46) | 0.001 | 0.004(0.004–0.02) | **0.001** |
| Nurse prescriber | 7(1.7) | 115(27.8) | 1 | | 1(omitted) | |
| Number of years in service | | | | | | |
| ≤5 years | 270(87.10) | 40(12.90) | 1 | | 1 | |
| >5 years | 11(10.58) | 93(89.42) | 0.81(0.76–0.87) | 0.000 | 0.69(0.60–0.78) | **0.000** |
| Training | | | | | | |
| Not trained | 14(3.4) | 28(6.7) | 1 | | | |
| Trained | 267(64.5) | 105(25.4) | 5.09(2.58–10.04) | 0.000 | 99.33(19.53–505.13) | **0.000** |
| Last training | | | | | | |
| ≤6 months | 248(60.0) | 118(28.5) | 1 | | 1 | |
| >6 months | 29(7.0) | 15(3.6) | 4.78(2.93–7.78) | 0.001 | 7.43(4.62–11.95) | **0.000** |

Nigeria 49% [28], Mali 50% [29]. In Ghana studies have shown the following findings, Greater Accra Region recorded that only 40% of the patients were tested for malaria [9]. In primary health care facilities in the middle belt of Ghana, there was 39.8% of malaria testing [19], Bongo District showed 91.2% [8]. Malaria testing in children under five years in rural Ghana was 98.2% [30] and 58.5% in Ho Municipality, Volta Region [18]. This variation in the testing

rate of suspected malaria cases could be attributed to the difference in the settings and availability of testing facilities. Studies have shown that in rural settings, testing rates were higher compared to urban settings [8, 30]. This could be because RDTs were readily available in rural facilities since microscopic services were scarce in those facilities. Also, RDTs takes less than 20 minutes to be completed unlike the longer time spent to conduct a microscopic test. The longer waiting time associated with receiving laboratory results for microscopy might have contributed to the lower levels of testing in urban settings. Our setting; Mfantseman Municipality is mostly peri-urban and the health facilities where our study was conducted provide microscopic services and this is most likely to have contributed to the lower testing rate in our study relating to the longer waiting time for microscopy. High rates of presumptive treatment without testing could lead to over-prescription of antimalarials which can in turn increase the morbidity and mortality associated with the disease. In order to prevent presumptive treatment and improve adherence to test-based management, patients and caregivers should be educated about the T3 strategy so that they can insist on being tested and only be prescribed antimalarials when tests are positive.

With regards to malaria treatment, findings from this study indicated that febrile patients were prescribed antimalarials, irrespective of whether they were tested or not, and whether the result was positive or negative for those tested. Prescribers prescribed ACTs for 93.2% patients. Comparing our findings to other studies, different levels of adherence to the recommended national treatment guidelines have been reported from 44% in Ogun State of Nigeria to 67.1% in Malawi [8, 16, 19, 26]. Contrary to our findings, higher adherence levels to the national treatment guidelines were observed in Kenya, Nigeria and Ghana [8, 16, 26]. The varying percentages of prescriber adherence to the recommended national treatment guideline can possibly be attributed to the availability of the testing facilities and recommended antimalarials in health facilities.

Prescribers in our study were more likely to prescribe treatment based on their clinical intuition without testing. This could be attributed to some experiences that has been gathered over time where all fevers were presumed to be caused by malaria and possibly a lack of understanding on the need to adhere to the T3 strategy. The treatment pattern in this study showed that more than half of febrile patients were not tested before treatment administration and for about a quarter of patients who tested negative, prescribers still treated all with antimalarials, these patients can be said to be inappropriately treated. Previous studies in the Greater Accra region and the middle belt of Ghana reported higher rates of over treatment of suspected malaria cases with ACT's [9, 19]. The overwhelming presumptive treatment seen in these previous studies could also be attributed to differences in the populations in those studies (children under 5 years) who are considered to be more susceptible to malaria infection.

Tracking of patients treated for malaria is essential in malaria control efforts because it helps in determining treatment outcomes. Findings from our study indicated that overall, over sixty percent of the patients treated were tracked i.e.; asking treated patients to return to the health facility for follow-up/review. Comparably, low adherence to tracking have been reported in Bongo and Ho districts in Ghana [8, 19]. However, higher percentage of tracking of patients after treatment was realized in a study by Ramseyer Ahmed at Atebubu-Amanten District, Ghana, where over ninety percent of all patients treated for malaria were followed-up/reviewed [31]. Prescribers will have to see the need to ensure follow-up on all cases of malaria they have treated in order to have an idea about the patient's treatment outcome.

Overall, findings from this study showed that about a third of febrile patients were treated according to the test, treat, and track strategy for malaria control in the Mfantseman Municipality. The low achievement of the percentage was mainly due to the lower rates of testing of febrile patient by the prescribers in our study. A similar study conducted among children

under 5 years in the Bongo District of Ghana observed overall compliance with the T3 strategy to be 42.5% [8], which is a slightly higher percentage adherence compared to findings from our study. Comparatively, higher percentage of adherence in Bongo District can be attributed to the fact that their study was among children under 5 years in whom the prevalence of malaria is high.

Factors associated with adherence to the T3 strategy were considered at health facility-level, prescriber-level and patient-level. Specifically, regarding health facility-level factors, malaria diagnostic capacity of the health facilities is required for malaria testing and their role in increasing adherence to the T3 strategy. Considering the increased odds of adherence among prescribers in facilities with more than three laboratory scientists compared to less than three laboratory scientists in our study, it can be deduced that the work load in the laboratory and the fewer numbers of scientists can hinder the testing rate of patients coupled with the longer time needed to complete microscopic investigations. Comparable conclusions were made by Bawate et al. in Uganda [6] and Bonful et al. in the Greater Accra Region of Ghana [9]. Conscious efforts must be made towards increasing the proportion of febrile patients being tested through RDT as well as reducing the waiting time associated with requesting and waiting for malaria test results from the laboratory.

Concerning prescriber-level factors, prescriber adherence decreased as their age increased. Their odds of adherence also decreased as the number of their years in service increased. This may be partly attributed to the fact that older prescribers by somewhat longer periods in service may have gained more experience towards symptomatic management. However, this association with age must be interpreted with caution because we do not know whether the factor was causally related to practice or simply a marker for another characteristic not measured in this study.

We also found higher levels off adherence among female prescribers compared to male prescribers. No previous literature has found higher levels of adherence among prescribers, it is therefore important to pay more attention to male prescribers when implementing interventions towards adherence.

Among medical officers, physician assistants, and nurse prescribers, medical officers were found to be more likely to adhere to the T3 strategy compared to physician assistants and nurse prescribers. Conversely to our observation, several studies have established lower adherence among medical officers compared to other categories of prescribers [6, 9, 19, 24, 32].

Furthermore, we found that prescribers who were trained on T3 strategy adhered to the strategy compared to those who have not been trained. Similarly, couple of similar studies have also shown an association of adherence with in-service training [26, 32]. However, another study established that training on malaria case management did not influence prescriber adherence to malaria treatment protocols [9]. Even among prescribers who have been trained, those whose last training was within six months had lower adherence levels compared to those training was over six months. The variation in findings from studies with regards to adherence among different professional levels can only be attributed to the fact that apart from professional levels and training on the T3 strategy of malaria control, individual attitudes towards adherence may also vary. Regardless of knowledge on the T3 strategy, an individual prescriber may either decide to adhere or not to adhere to the strategy. However, it remains important that prescribers are trained appropriately in the T3 strategy.

The lower odds of adherence among prescribers who have been monitored by NMCP staff is difficult to understand. One would have thought that monitoring would rather contribute to prescriber adherence, however the opposite is the case in our study. Nevertheless, there is the need for continuous and more intensified monitoring and supportive supervision in order to ensure adherence to the T3 strategy in the municipality as established by some studies [9, 33].

In relation to the patient-level factors associated with adherence, we observed that prescribers were more likely to adhere to the strategy when managing patients aged five to sixty-four years compared to children less than five years. Contrary to our finding, Kwarteng et al found high prescriber adherence; 61%, when managing children under five years [19] in the Volta Region. It is however worrying that our study showed low levels of adherence towards children under five years. High level of prescriber adherence is necessary when managing malaria in children under 5 years due to the fact that the prevalence of the disease is higher in children compared to adults. Additionally, children are considered a high-risk group for malaria and can have various differential diagnosis for fever. With the country investing majorly in reduction in under 5 mortalities due to all causes, it is in line that prescribers should be more cautious hence adhering to T3 more when children are concerned.

## Study limitations and strength

Electronic data on febrile out-patients was mainly secondary data that were collected without gold standard and this implies that any probable misclassification could not be reported. Prescribers were interviewed at their respective places of work, well aware of the biases, it cannot be guaranteed that respondents did not alter their responses despite their true experiences and practices. Because our study was a retrospective one, authors did not consider using each of the components of T3 as an individual outcome and did not determine the corresponding exposure variables. It will be very easy to take this into consideration when conducting a prospective study. The data set used to reach these results is cross-sectional and discovered associations between independent and dependent variables cannot be assumed to be causal. Hence, one should be careful in interpreting findings from this study. Nevertheless, the strength of this study rests in the rigorous quantitative methods and validation of findings from the secondary data using interviews to assess prescriber adherence to the T3 strategy for malaria control in the Mfantseman Municipality.

## Conclusions and recommendations

Adherence to the T3 strategy in Hospitals is low in the Mfantseman Municipality of the Central Region of Ghana. The factors there were associated adherence were age of the patient, number of qualified laboratory scientists and monitoring at patient and health facility levels. Additionally, prescriber related factors including age, sex, professional category, number of years in service and training were associated with adherence to the T3 strategy for malaria control.

We recommend that the NMCP should deploy quality improvement tools during the supervision of health workers and rely on less didactic training strategies but more on the job training, that could help improve outcomes after training. Health facilities should consider performing RDTs for febrile patients at the OPD during the triaging process in order to minimize patient waiting time associated with requesting and waiting for malaria test results from the laboratory. Facility heads should also place a priority on low cadre prescribers as well as male prescribers during the planning and implementation of interventions to improve adherence to the T3 strategy.

## Recommendation for future research

This study was carried out in two major referral hospitals in the Mfantseman Municipality. It is however recommended that other researchers look at carrying out similar studies at other levels of the health care system such as Polyclinics, Clinics, Health Centers and CHPS compounds in the municipality to also determine their level of adherence to the T3 strategy of

malaria control. Future researchers should also consider treating each individual component of the T3 strategy as outcomes to determine how appropriate exposure variables can influence adherence.

## Supporting information

**S1 Dataset. Manuscript dataset (T3_dataset. excel).**
(XLS)

## Acknowledgments

The authors would like to acknowledge the Management of the Central Regional Health Directorate, Mfantseman Municipal Health Directorate, Management and staff of Saltpond Municipal and Mercy Women's Catholic Hospital most especially Abdu Aziz Mudasiru and Rashid Ntiamoah the research assistants for this project.

## Author Contributions

**Conceptualization:** Ernestina Esinam Agbemafle, Chrysantus Kubio, Donne Kofi Ameme.

**Data curation:** Ernestina Esinam Agbemafle, Harriet Affran Bonful.

**Formal analysis:** Ernestina Esinam Agbemafle, Magdalene Akos Odikro, Harriet Affran Bonful.

**Methodology:** Ernestina Esinam Agbemafle, Adolphina Addo-Lartey, Harriet Affran Bonful.

**Project administration:** Ernestina Esinam Agbemafle, Joseph Asamoah Frimpong, Donne Kofi Ameme.

**Supervision:** Adolphina Addo-Lartey, Samuel Oko Sackey, Harriet Affran Bonful.

**Validation:** Ernestina Esinam Agbemafle, Harriet Affran Bonful.

**Visualization:** Ernestina Esinam Agbemafle, Magdalene Akos Odikro.

**Writing – original draft:** Ernestina Esinam Agbemafle.

**Writing – review & editing:** Ernestina Esinam Agbemafle, Adolphina Addo-Lartey, Magdalene Akos Odikro, Joseph Asamoah Frimpong, Chrysantus Kubio, Donne Kofi Ameme, Samuel Oko Sackey, Harriet Affran Bonful.

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
