## [Decision Letter · Decision Letter 0]

4 Oct 2022

PONE-D-22-22129Adherence to the Test, Treat and Track Strategy for Malaria Control Among Prescribers, Mfantseman Municipality, Central Region, GhanaPLOS ONE

Dear Dr. Odikro,

Thank you for submitting your manuscript to PLOS ONE. After careful consideration, we feel that it has merit but does not fully meet PLOS ONE’s publication criteria as it currently stands. Therefore, we invite you to submit a revised version of the manuscript that addresses the points raised during the review process.

We look forward to receiving your revised manuscript.

Kind regards,

Anupkumar R. Anvikar, M.D.

Academic Editor

PLOS ONE

Journal Requirements:

2. In the ethics statement in the manuscript and in the online submission form, please provide additional information about the patient records/samples used in your retrospective study. Specifically, please ensure that you have discussed whether all data/samples were fully anonymized before you accessed them and/or whether the IRB or ethics committee waived the requirement for informed consent. If patients provided informed written consent to have data/samples from their medical records used in research, please include this information.

4. Please amend the manuscript submission data (via Edit Submission) to include author Ernestina Esinam Agbemafle and Adolphina Addo-Lartey. 

6. We note you have included a table to which you do not refer in the text of your manuscript. Please ensure that you refer to Table 2 in your text; if accepted, production will need this reference to link the reader to the Table.

Reviewers' comments:

Reviewer's Responses to Questions

**Comments to the Author**

1. Is the manuscript technically sound, and do the data support the conclusions?

Reviewer #1: Yes

Reviewer #2: Partly

Reviewer #3: Yes

2. Has the statistical analysis been performed appropriately and rigorously? 

Reviewer #1: Yes

Reviewer #2: Yes

Reviewer #3: Yes

3. Have the authors made all data underlying the findings in their manuscript fully available?

Reviewer #1: No

Reviewer #2: Yes

Reviewer #3: Yes

4. Is the manuscript presented in an intelligible fashion and written in standard English?

Reviewer #1: Yes

Reviewer #2: Yes

Reviewer #3: Yes

5. Review Comments to the Author

Reviewer #1: Introduction:

The main aim of the study is “to assess the level of adherence to the T3 strategy”. It appears the assessment has been done at two distinct levels (1) Identify the outcome (by review of patient records) (2) Identify the exposure variables (an important component is interview of the prescribers). Since the methods of assessment of these two are completely different, they may be mentioned separately. For example, there is confusion about study population and study subjects (the study subjects should come from the study population).

Methods:

Study population (lines 121-127): A brief description about the health care system (where does the patient go first? Is it free or on payment? Who is authorized to treat? What is the routine method for diagnosis of malaria? Who were the prescribers? How follow up is enforced? Etc). Some background information regarding educational background and health care responsibility of medical officers, physician assistants, and nurse prescribers may be mentioned.

Sample Size: The necessity of sampling will be realized only when the total number of records available for analysis is known. It will be easier for the readers to understand if the values of each component of the formula for sample size calculation is mentioned (e.g, there is no mention of the confidence level.)

Sampling: What was the justification of using systematic random sampling instead of simple random sampling? What was the interval?

Outcome: Each individual component of the outcome (T3) can be treated as independent, because they are influenced by different predictors. For example testing will depend on the availability of the RDT and laboratory facilities, while treating will depend on availability of antimalarial drugs. Therefore, individual components of T3 can be treated as outcome and appropriate exposure variables may be selected accordingly.

Exposure variables: Since the prescribers belong to three different categories (e.g, medical officers, physician assistants, and nurse prescribers), whose professional competence may vary widely, the influence of these categories on the outcome can be studied separately. Since, predictor variables have not been defined clearly, all sorts of associations have been tried.

Discussion:

Discussion has concentrated more on the comparison with other studies but less on the possible causes of failure.

Reviewer #2: The manuscript entitled "Adherence to the Test, Treat and Track Strategy for Malaria Control Among

Prescribers, Mfantseman Municipality, Central Region, Ghana" submitted for publication mainly analysed the data retrospectively on 414 febrile patients attended the Saltpond Municipal Hospital and Mercy Women’s Catholic Hospitals for treatment. Of them only 180 (43.5%) were tested with 138 (76.7%) testing positive. But all have been administered with antimalarials. Of the total cases only 127 (30.7%) were adhered to T3 strategy. The information is of regional importance .

Comments

1. The introduction section should highlight the importance of the study from programme point of view.

2. The authors should check the discussion section because many from line no 258 to 310 missing and 310 -326 wrongly uploaded.

Reviewer #3: Comments

The research article entitled “Adherence to the Test, Treat and Track (T3) Strategy for Malaria Control among prescribers of Mfantseman Municipality, Central Region, Ghana” submitted by Agbemafle and others have analysed the outpatient records of 414 febrile patients, attended Saltpond Municipal Hospital and Mercy Women’s Catholic Hospitals in Mfantseman Municipality, Ghana in 2020. The authors attempted mainly to find out the rate of adherence to T3 strategy introduced for malaria elimination by the WHO in 2010. The authors have found that of the out of total febrile cases, 43.5% were tested with 76.7% test positivity rate, but all febrile cases have been administered with antimalarials. Of them only 30.7% have adhered to T3 strategy. This is a very important observation from programme point of view.

However, the following comments may be looked at before publication.

1. The introduction section has not clearly mentioned the objective/importance of the present study

2. The method section needs improvement. The authors should have mentioned only the essential and critical steps in different sub headings

3. In the result section the manufacturer of RDTs used for diagnosis and the partner drug of ACT administered should be clearly mentioned.

4.The caption of Table 2 should be precise.

5. The discussion section is missing many of the sentences from line 258- 326.

Recommendation: The data presented in the manuscript bears regional importance and hence most suitable for regional publication.

Other areas attempting for maximum adherence to T3 (which is essential for malaria elimination) may take clues, if published.

6. PLOS authors have the option to publish the peer review history of their article (what does this mean?). If published, this will include your full peer review and any attached files.

Reviewer #1: No

Reviewer #2: No

Reviewer #3: **Yes: **Madan Mohan Pradhan

---

## [Author Response · Author response to Decision Letter 0]

25 Nov 2022

Manuscript ID: PONE-D-22-22129; 

Manuscript Title: Adherence to the Test, Treat and Track Strategy for Malaria Control Among Prescribers, Mfantseman Municipality, Central Region, Ghana 

POINT BY POINT RESPONSE TO COMMENTS

Journal Requirements:

Response: We have formatted the manuscript in line with PLOS ONE journal requirements

2. In the ethics statement in the manuscript and in the online submission form, please provide additional information about the patient records/samples used in your retrospective study. Specifically, please ensure that you have discussed whether all data/samples were fully anonymized before you accessed them and/or whether the IRB or ethics committee waived the requirement for informed consent. If patients provided informed written consent to have data/samples from their medical records used in research, please include this information……………………..

Response: all data set of patients were fully anonymized before we accessed the data set. This has been clearly stated in the ethics section in line 223-224

Response: There are no ethical restrictions with regards to the data set used in this study, we have submitted the data set along with this manuscript

4. Please amend the manuscript submission data (via Edit Submission) to include author Ernestina Esinam Agbemafle and Adolphina Addo-Lartey. 

Response: Thank you, we have updated the manuscript submission system.

Response: Figure 1 was drawn by authors using the Quantum Geographical information System (QGIS) and has no issues relating to copywrite. However, we have described the study site extensively and deleted the figure completely from the manuscript. 

6. We note you have included a table to which you do not refer in the text of your manuscript. Please ensure that you refer to Table 2 in your text; if accepted, production will need this reference to link the reader to the Table.

Response: Table 2 has been appropriately referred to in line 278-289

Reviewers' comments

Comments to the Author

Reviewer#1:

Introduction:

1. The main aim of the study is “to assess the level of adherence to the T3 strategy”. It appears the assessment has been done at two distinct levels (1) Identify the outcome (by review of patient records) (2) Identify the exposure variables (an important component is interview of the prescribers). Since the methods of assessment of these two are completely different, they may be mentioned separately. For example, there is confusion about study population and study subjects (the study subjects should come from the study population).

Response: The two distinct methods used in the assessment have been described under study design as suggested and the corrections have been made regarding study population and study subjects (lines 111-115)

Methods:

2. Study population (lines 121-127): A brief description about the health care system (where does the patient go first? Is it free or on payment? Who is authorized to treat? What is the routine method for diagnosis of malaria? Who were the prescribers? How follow up is enforced? Etc). Some background information regarding educational background and health care responsibility of medical officers, physician assistants, and nurse prescribers may be mentioned.

Response: A brief description of the healthcare system and background information about prescribers have been provided under the operation of the health care system (lines 136-158)

3. Sample Size: The necessity of sampling will be realized only when the total number of records available for analysis is known. It will be easier for the readers to understand if the values of each component of the formula for sample size calculation is mentioned (e.g, there is no mention of the confidence level.)

Response: The total number of record available for analysis was earlier mentioned under sampling, however, for clarity, sampling and sample size calculation has been merged to provide a good flow of information (lines 163-180). Also, the terms in the formula for sample size calculation have been described and 95% confidence interval specified (160-181)

4. Sampling: What was the justification of using systematic random sampling instead of simple random sampling? What was the interval? 

Response: Thank you for pointing this out. Due to the large number of datasets, we used systematic sampling to ensure that the data selected is spread across the patients seen at all hours of each day since facilities provide 24/7 care and the electronic database captures as such. Sampling interval was calculated for each site based on their average daily number of cases and used (interval of 2 for both sites). This has been explained further in the write-up (Lines175-185)

5. Outcome: Each individual component of the outcome (T3) can be treated as independent, because they are influenced by different predictors. For example testing will depend on the availability of the RDT and laboratory facilities, while treating will depend on availability of antimalarial drugs. Therefore, individual components of T3 can be treated as outcome and appropriate exposure variables may be selected accordingly.

Response: Thank you for this observation. it is very true that each of the component of the outcome (T3) can be treated as independent, however, authors did not consider using each of the components as an individual outcome and did not determine the corresponding exposure variable. Ours was a retrospective study; it will be very easy to take this into consideration when conducting a prospective study. We have therefore recommended this to be considered in future studies and also added it to the limitations of the study (lines 430-432 and 456-458).

6. Exposure variables: Since the prescribers belong to three different categories (e.g, medical officers, physician assistants, and nurse prescribers), whose professional competence may vary widely, the influence of these categories on the outcome can be studied separately. Since, predictor variables have not been defined clearly, all sorts of associations have been tried.

Response it is very true that prescribers belong to different categories whose competencies vary widely, however the authors decided to combine these professionals because the study was conducted only in the Mfantseman municipality and considering only particular groups will result in smaller sample size. Moreso, training of prescribers on the T3 strategy is a common variable that is likely to influence adherence no matter the professional category of the prescriber. We instead considered the categories as a factor that may be associated with adherence to the T3 strategy ( Table 2)

Discussion:

7. Discussion has concentrated more on the comparison with other studies but less on the possible causes of failure.

Response: Thank you for this observation. We have described possible causes of failure; examples include mentioning of the possibility of non-adherence to the testing component of the strategy in the hospitals being the longer waiting time associated with receiving laboratory results. The longer wait period might have contributed to the lower levels of testing in urban settings and also facilities that offer microscopy testing service. (Lines 314-423)

With regards to the treatment patterns, authors attributed this to the availability of the recommended antimalarials in health facilities; prescribers are more likely to prescribe treatment based on the availability of ACTs in the health facility (line 319-323)

Reviewer 2

Comments

1. The introduction section should highlight the importance of the study from programme point of view.

Response: This has been addressed in the introduction section (lines 68-86)

2. The authors should check the discussion section because many from line no 258 to 310 missing and 310 -326 wrongly uploaded.

Response: Thank you, all missing sentences in this section have been inserted (lines 296-423)

Reviewer 3

1. The introduction section has not clearly mentioned the objective/importance of the present study

Response: the objectives of the study have been clearly mentioned in the introduction (Lines 106 to 108)

2. The method section needs improvement. The authors should have mentioned only the essential and critical steps in different sub headings

Response: this is well noted. We have revised and mentioned critical steps to ensure good flow of information in the methods section (lines 110-231)

3. In the result section the manufacturer of RDTs used for diagnosis and the partner drug of ACT administered should be clearly mentioned.

Response: Thank you for this suggestion. However, authors maintained the generic names of the drugs and used RDT’s in general because various ACTs are administered in health facilities in Ghana. Since our focus was not on the efficacy of the drug or the RDT’s but on whether the T3 strategy was adhered to, we chose to omit this information for the drugs and RDT’s as it was not collected as part of our retrospective study. 

4.The caption of Table 2 should be precise.

Response: The table heading has been precisely labelled (lines 284 -285)

5. The discussion section is missing many of the sentences from line 258- 326.

Response: Thank you, all missing sentences in this section have been inserted and modified from (line 299-426)

---

## [Editor Report · Decision Letter 1]

13 Dec 2022

Adherence to the Test, Treat and Track Strategy for Malaria Control Among Prescribers, Mfantseman Municipality, Central Region, Ghana

PONE-D-22-22129R1

Dear Dr. Odikro,

We’re pleased to inform you that your manuscript has been judged scientifically suitable for publication and will be formally accepted for publication once it meets all outstanding technical requirements.

Kind regards,

Anupkumar R. Anvikar, M.D.

Academic Editor

PLOS ONE
---

## [Editor Report · Acceptance letter]

21 Dec 2022

PONE-D-22-22129R1 

Adherence to the Test, Treat and Track Strategy for Malaria Control Among Prescribers, Mfantseman Municipality, Central Region, Ghana 

Dear Dr. Odikro:

I'm pleased to inform you that your manuscript has been deemed suitable for publication in PLOS ONE. Congratulations! Your manuscript is now with our production department. 

Kind regards, 

on behalf of

Dr. Anupkumar R. Anvikar 

Academic Editor

PLOS ONE